# Identification of Methicillin-Resistant *Staphylococcus aureus* (MRSA) Genetic Factors Involved in Human Endothelial Cells Damage, an Important Phenotype Correlated with Persistent Endovascular Infection

**DOI:** 10.3390/antibiotics11030316

**Published:** 2022-02-26

**Authors:** Xia Xiao, Yi Li, Liang Li, Yan Q. Xiong

**Affiliations:** 1The Lundquist Institute for Biomedical Innovation at Harbor-UCLA Medical Center, Torrance, CA 90502, USA; xiaoxia@yzu.edu.cn (X.X.); yi.li@lundquist.org (Y.L.); bfcmla@gmail.com (L.L.); 2College of Veterinary Medicine, Yangzhou University, Yangzhou 225009, China; 3Center for Devices and Radiological Health, U.S. Food and Drug Administration, Silver Spring, MD 20993, USA; 4David Geffen School of Medicine at UCLA, Los Angeles, CA 90095, USA

**Keywords:** MRSA, human endothelial cell damage, virulence factors

## Abstract

Methicillin-resistant *Staphylococcus aureus* (MRSA) is a leading cause of life-threatening endovascular infections. Endothelial cell (EC) damage is a key factor in the pathogenesis of these syndromes. However, genetic factors related to the EC damage have not been well studied. This study aims to identify genetic determinants that impact human EC damage by screening the genome-wide Nebraska Transposon Mutant Library (NTML). A well-established MTT assay was used to test the in vitro damage of human EC cell line (HMEC-1) caused by each mutant strain in the NTML. We first confirmed some global regulators and genes positively impact the EC damage, which is consistent with published results. These data support the utility of the high-throughput approach. Importantly, we demonstrated 317 mutants significantly decreased the EC damage, while only 6 mutants enhanced the EC damage vs. parental JE2 strain. The majority of these genes have not been previously defined to affect human EC damage. Interestingly, many of these newly identified genes are involved in metabolism, genetic and environmental information processing, and cellular processes. These results advance our knowledge of staphylococcal genetic factors related to human EC damage which may provide novel targets for the development of effective agents against MRSA endovascular infection.

## 1. Introduction

*Staphylococcus aureus* is the most common cause of endovascular infection, including infective endocarditis (IE). Despite the use of gold-standard antibiotics, morbidity and mortality associated with these syndromes remain unacceptably high [1]. In addition, the emergence of methicillin-resistant *S. aureus* (MRSA) further complicates the management of patients with these infections and emphasizes this public health threat [1]. Therefore, there is an urgent need to understand specific genetic factors involved in the pathogenesis and antibiotic treatment outcome of MRSA endovascular infection.

It is generally recognized that the pathogenesis of *S. aureus* is complex and probably involves the coordinate expression of multiple gene products, including a variety of surface adhesive proteins and exoproteins [2]. Once *S. aureus* enters into the bloodstream, it must avoid host innate defense killing to survive. When the organism has persisted in the bloodstream, it must then colonize and invade the endothelial cells (ECs) lining of the blood vessels, and, subsequently, damage the ECs to infect deeper tissues to cause organ dissemination [3]. It has been well demonstrated that EC damage plays a crucial role in the pathogenesis of many human diseases, including endovascular infections [4]. In addition, we have recently demonstrated a positive correlation between in vitro human EC damage and virulence, as well as vancomycin treatment persistent outcome in an experimental endocarditis model caused by clinical MRSA isolates [5]. However, little is known about the genetic factors involved in the EC damage in *S. aureus*. 

The Nebraska Transposon Mutant Library (NTML) consists of 1920 sequence-defined transposon insertion mutants of non-essential genes in a community-associated (CA) MRSA USA300 strain, JE2 [6]. This library has been used for screening several biological phenotypes, including hemolysis, proteolysis, carotenoid pigment formation, antibiotic susceptibility, and biofilm formation [6,7,8]. These investigations demonstrate that the NTML may serve as a valuable genetic tool to study host-pathogen interaction.

Numerous investigations have used human umbilical vein EC (HUVECs) to study microbial–EC interactions. However, the use of HUVECs requires a constant supply of umbilical cords, and there are significant donor-to-donor variations in these ECs. To overcome these difficulties, immortalized ECs, including human microvascular EC (HMEC-1), have been developed. These cell lines have better availability and less variability [9]. In addition, we previously compared *S. aureus* EC damage with HMEC-1 cell line and HUVECs, and found HMEC-1 cells were more susceptible to damage caused by *S. aureus* vs. HUVECs [10]. In addition, the HMEC-1 cell line has been used to study the EC interactions with multiple microorganisms, including *S. aureus* [10,11,12]. Thus, in the current investigation, the HMEC-1 cell line was employed to test the impact of all the mutant strains in the NTML on its damage.

In the current study, we aimed to identify staphylococcal genes associated with the EC damage by performing an unbiased genome-wide screening of all mutations in the NTML. This study will remarkably advance our understanding of staphylococcal genetic factors related to human EC damage which may provide novel targets for the development of effective compounds against MRSA endovascular infections.

## 2. Results

### 2.1. The MTT Assay Is Applicable to the High Throughput Screening of Genes Involved in HMEC-1 Damage

We confirmed some *S. aureus* genetic factors which have previously been reported to affect EC damage. For instance, global regulator (e.g., *agr*, *saeSR*, and *arlSR*) and structural genes related to gamma-hemolysin (e.g., *hlg*) and serine-like protease (e.g., *spl*) positively impact EC damage. In addition, the control *arlR* mutant strain caused significantly less EC damage (<30%) vs. JE2 parental strain, which is in accordance with the previously reported results. These results proved the feasibility and reliability of this high throughput screening assay.

### 2.2. Identified Staphylococcal Genes Impacting HMEC-1 Damages

The mean HMEC-1 damage rate caused by the JE2 parental strain is 46.19 ± 2.97%. To focus on the genes which highly affect the EC damage, we set up the EC damage rates of ≤30% or ≥60% with *p* values less than 0.05 as cutoffs for data analysis. Screening of the whole NTML displayed that 317 individual gene mutations led to significantly decreased HMEC-1 damage rates (≤30%; *p* <0.05; Figure 1, Table 1), suggesting these genes positively impact the EC damage. Only six mutant strains demonstrated significantly increased HMEC-1 damage (≥60%, *p* < 0.05; Figure 1, Table 2), including four genes with known functions (e.g., *mepA*, *azoR*, and *moaD*, and SAUSA300_1197) and two hypothetical genes with unknown function. EC damage rates of the rest mutants from the NTML were presented in Appendix A. JE2 parental strain and randomly selected mutants showed similar EC damage rates between 24-well and 384-well plates assay (Table 3). Some of the mutants that caused significant changes to EC damage were successfully classified into KEGG categories, including metabolism, genetic information processing, environmental information processing, and cellular processes (Table 4). For the KEGG categories, ~65% of genes functioned in metabolism pathways, ~24% involved in environmental information processing, ~11% acted in genetic information processes, and ~9% associated with cellular processes (Figure 2). In addition, some of these genes had multiple functions in the different KEGG pathways.

## 3. Discussion

It is well recognized that EC damage plays a crucial role in the pathogenesis of *S. aureus* endovascular infection [5,13,14]. For instance, we have demonstrated a positive correlation between in vitro EC damage and virulence, as well as antibiotic treatment persistent outcome in an experimental endocarditis model caused by clinical MRSA isolates [5]. In addition, we also noticed that clinical MRSA strains collected from patients with persistent bacteremia cause significantly greater EC damage compared to clinical resolving MRSA isolates [15]. Moreover, the inactivation of *agr*, *saeR*, and *arlSR* has been proved significantly reduce EC damage as compared to their respective parental strains [13,16]. However, these studies only focused on a few virulence factors in *S. aureus*. Thus, the current study was designed to broadly define genetic determiners in *S. aureus* which involve in human EC damage using a high-throughput approach to screen a transposon mutant library containing 1920 non-essential gene mutants in MRSA USA300 JE2 background.

In the current study, we first verified the reliability of our high-throughput screening system. Consistent with previous reports [13,16], we demonstrated that the inactivation of global regulators such as *agr*, *arlRS*, or *saeRS* significantly decreases EC damage. In addition, consistent results were obtained between 384-well and 24-well plates assays, which validated the improvement of testing significantly more samples each time.

Several interesting and important observations emerged from the present investigations. Overall, over 320 mutants had a significant impact on the EC damage. The majority of these mutants significantly reduced EC damage vs. JE2 parental strain. Using KEGG pathway analysis, mutant strains were classified into four categories, including metabolism, genetic information processing, environmental information processing, and cellular processes (Figure 3). Only six mutants were found with significantly increased EC damage vs. JE2 parental strain. Importantly, many of these genes are not previously defined to impact human EC damage in *S. aureus*.

Many staphylococcal genetic factors related to metabolism were shown to intimately impact the EC damage. For instance, several gene mutants related to carbohydrate metabolism including tricarboxylic acid (TCA) cycle (e.g., *pdhA*, and *lpdA*) showed significantly decreased EC damage. Inactivation of *pdhA* or *lpdA* was reported to be associated with slower growth [17,18]. Since the TCA cycle processes produce the main energy resources for cellular activities [19], inactivation of corresponding TCA genes may result in lack of energy which may subsequently cause slower growth and decrease EC damage. In addition, mutants with genes related to energy metabolism (e.g., *cyoE*, and *atpH*) also displayed lower EC damage rates vs. parental strain JE2. It has been reported that *cyoE* encoding a protoheme IX farnesyltransferase is essential for processing heme into the electron transport chain and plays a critical role in cytolytic toxins production in *S. aureus*. Deletion of *cyoE* in *S. aureus* significantly decreases the expression of cytolytic toxins [20]. Turner et al. reported that mutation of *aptH* (associated with ATP synthase) had attenuated virulence and less invasiveness in vivo [21]. These results suggest that genetic factors associated with energy metabolism have activities on EC damage that may link to virulence. Lipid metabolism genes (e.g., *gehB*, and *ugtP*) were reported to promote biofilm formation and host cell invasion [22]. We found that the mutation of these genes had significantly decreased EC damage vs. JE2 parental strain. These results may indicate a connection between lipid metabolism and EC damage. Genetic factors associated with nucleotides metabolism (e.g., *purN*) were also found to positively impact the EC damage. *purN* encodes the enzyme in *de novo* purine biosynthesis pathway which generates ATP and GTP that can be processed to stringent response alarmone, guanosine 3′-diphosphate-5-di(tri)phosphate ((p)ppGpp) [15]. Increased GTP and subsequent (p)ppGpp levels lead to enhanced persistent bacteremia (PB) phenotypes including a higher EC damage rate [15]. It is worthwhile to mention, genes related to staphylococcal cell-wall peptidoglycan biosynthesis (e.g., *murA*) and cell division (e.g., *scdA*) showed significant positive effects on EC damage. Cell-wall synthesis has long been considered an important target for novel anti-*S. aureus* agents [23,24], and our findings have implications for the approach.

In the genetic information processing pathways, genes involved in homologous recombination (e.g., *recD*, and *recG*), ribosome (e.g., *rrlA*, and *rpsA*), and protein export (e.g., *lspA*, and *tatA*) were identified to affect EC damage. For example, the signal peptidase encoded by *lspA* is required for biogenesis of bacterial lipoproteins, and failure to produce mature lipoproteins has previously been shown to impair pathogenicity and immune-modulating [25]. The results suggested that some genes related to genetic information processing also play a role in human EC damage.

The inactivation of genes involved in environmental information processing pathways such as ABC transporter (e.g., *fhuB*, and *mntC*) and two-component system (e.g., *saeSR*, and *arlSR*) also decreased EC damage. These findings were in accordance with previous studies showing the presence of these gene products was associated with higher in vivo virulence potential vs. their respective WT strains [13,26,27,28].

Genes involved in cellular process, specifically quorum sensing (e.g., *agr*, and *luxS*), were identified to contribute to the EC damage. It is well known that quorum sensing via *agr* plays a central role in the pathogenesis of *S. aureus*. Under high cell density, *agr* is responsible for the increased expression of many toxins which may impact the EC damage [16], while the function of *luxS* in *S. aureus* has not been well investigated. 

Genes unidentified in the KEGG pathways also showed a positive impact on the HMEC-1 damage in the current study. Some of these genes have been previously demonstrated to correlate with biofilm formation (e.g., *xerC*), oxidative killing (e.g., *nfu*, and *yjbI*), hemolysis (e.g., *hlb*), and heat shock (e.g., *hslU*) [29,30,31,32]. In addition, few phage genes (SAUSA300_1433, SAUSA300_1934, SAUSA300_1936, SAUSA300_1968) were also shown impacts on the HMEC-1 damage. 

Mutants of six genes had elevated EC damage indicating their negative impact on the EC damage. Among these genes, *mepA* encodes a multidrug efflux pump protein [33], *azoR* encodes quinone reductase [34], *moaD* encodes one of the subunits of molydopterin synthase involved in sulfur relay system pathway [35], gene SAUSA300_1197 encodes glutathione peroxidase. Further investigations related to the relationship between these genes and EC damage are needed. 

## 4. Materials and Methods

### 4.1. Bacteria and Growth Conditions

The strains used in the current study include MRSA JE2 (a plasmid-cured derivative of LAC USA300) and 1920 transposon non-essential gene mutants within the NTML [6]. The NTML was kindly provided by the Network on Antimicrobial Resistance in *Staphylococcus aureus* (NARSA). The library was supplied in five 384-well microtiter plates. The plates containing MRSA mutant strains were duplicated and cultured in tryptic soy broth (TSB; Becton, Dickinson and Company, Franklin Lakes, NJ, USA). On the experiment day, bacterial strains were freshly inoculated in TSB media and cultured at 37 °C for 3 h to obtain logarithmic phase cells [36], and adjusted to an OD_600nm_ of 0.500 (~10^8^ CFU/mL) and diluted accordingly. *S. aureus* inocula were confirmed by quantitative culture.

### 4.2. Endothelial Cell (HMEC-1) Culture

The HMEC-1 cell line was obtained from Kathryn Kellar, of the Centers for Disease Control (CDC), in the U.S., and maintained as recommended [10]. Primary cells were established from human dermal microvascular endothelial cells and immortalized by transfection with a Pbr322-based plasmid containing the coding region for the simian virus 40 large T-antigen [10]. 

### 4.3. HMEC-1 Damage Assay

The effect of MRSA strains on EC damage was determined using a well-established 3-(4,5-dimethylthiazol-2-yl)-2,5-diphenyltetrazolium bromide (MTT) assay as described previously [13,37,38]. Briefly, logarithmic phase MRSA cells (1 × 10^5^ CFU/well) were added to HMEC-1 cells in 384-well plates with a density of ~5 × 10^3^ EC/well in MCDB131 medium to reach a multiplicity of infection (MOI) of 20, which JE2 parental caused ~50% HMEC-1 damage as established in our pilot experiments. After 3 hr invasion, extracellular MRSA cells were killed by adding lysostaphin (10 μg/mL) in full medium MCDB131 (Sigma-Aldrich, St. Louis, MO, USA) supplemented with 20% bovine calf serum, 2 mM glutamine, 100 IU/mL penicillin, and 100 mg/mL streptomycin [13,37]. At 18 hr incubation at 37 °C, MTT (5 mg/mL; Sigma-Aldrich, St. Louis, MO, USA) in Hank’s Balanced Salt Solution (HBSS, Thermo Fisher Scientific, Waltham, MA, USA) was added and incubated for 2 h, then the medium was replaced with 0.04 M HCl in absolute isopropanol (Thermo Fisher Scientific, Waltham, MA, USA) to stop the reaction and lyse the cells. Absorbance was measured at 560 nm (A_560nm_) using a microplate reader Synergy 2 (BioTek, Winooski, VT, USA). Uninfected HMEC-1 served as a negative control, and wells containing medium alone were used for background correction in each round. In addition, EC infected with Δ*arlR* in JE2 was selected as an additional control group as it was reported that *arlSR* inactivation leads to >70% reduction in human EC damage vs. JE2 parental strain [13]. EC damage was calculated using the following formula: 1 − (A_560nm_ of test well/A_560nm_ of 0% − damage control well) as previously described [37]. Each experiment was performed three times in triplicate.

### 4.4. Verification of the HMEC-1 Damage Screening Results

After the screening of the whole library, JE2 WT strain and 20 randomly selected mutant strains with significantly decreased EC damage were confirmed again with the same MTT method using 24-well plates. In addition, the mutant strains with significantly increased EC damage were also tested in 24-well plates to confirm the damage results with the same method. 

### 4.5. Statistical Analysis

Statistical analysis was performed using GraphPad Prism 9 (GraphPad Software, Inc., San Diego, CA, USA). *p*-values were determined using the paired rank-sum test between mutant and JE2 wild-type strains. *p* < 0.05 was considered statistically significant.

### 4.6. KEGG Enrichment Analysis

The genes that caused a significant change in EC damage were classified using the Kyoto Encyclopedia of Genes and Genomes (KEGG) mapper tool with the mode of *Staphylococcus aureus* subsp. aureus USA300-FPR3757 (saa) [39]. The genes from different KEGG pathway categories were further analyzed. 

## 5. Conclusions

To our knowledge, the present study provides the first whole-genome screen to identify genetic factors that impact human EC damage in *S. aureus*. Importantly, we defined a set of staphylococcal genes, which are not previously known to be associated with EC damage, significantly contribute to this phenotype. Although these findings need to be further verified using mutation strains generated by gene deletion and complementation techniques, our results provide new insights into the relationship between genetic factors and EC damage in *S. aureus*. These genetic factors may be ideal targets for the development of effective therapeutic strategies to treat invasive MRSA endovascular infection.

## Figures and Tables

**Figure 1 antibiotics-11-00316-f001:**
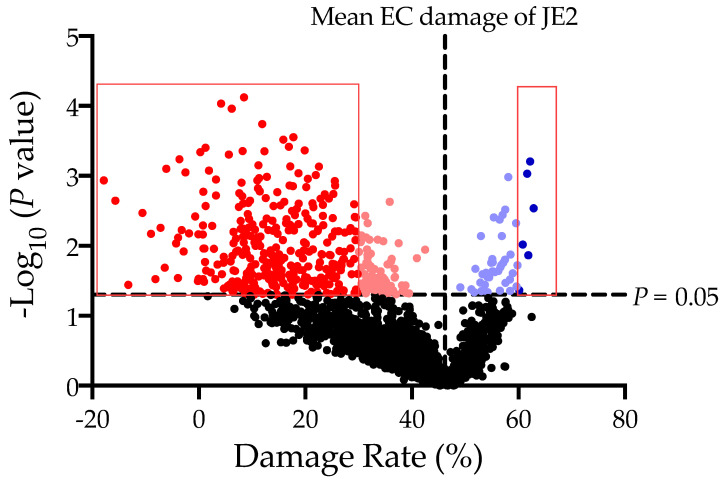
The global map of in vitro HMEC-1 damage rate caused by the mutant strains in the NTML. The vertical dashed line represents the mean of HMEC-1 damage rate of parental strain USA300 JE2 (46.19%); and the horizontal dashed line represents the *p* value of 0.05. The bright red dots represent ≤30% EC damage caused, while the bright blue dots represent ≥60% EC damage due to the study mutant strains in the NTML and *p* < 0.05 vs. JE2 WT strain. Damage rate below zero means the A_560nm_ of the test well is higher than the A_560nm_ of the negative damage control, which indicates that the mutant causes no damage to the EC.

**Figure 2 antibiotics-11-00316-f002:**
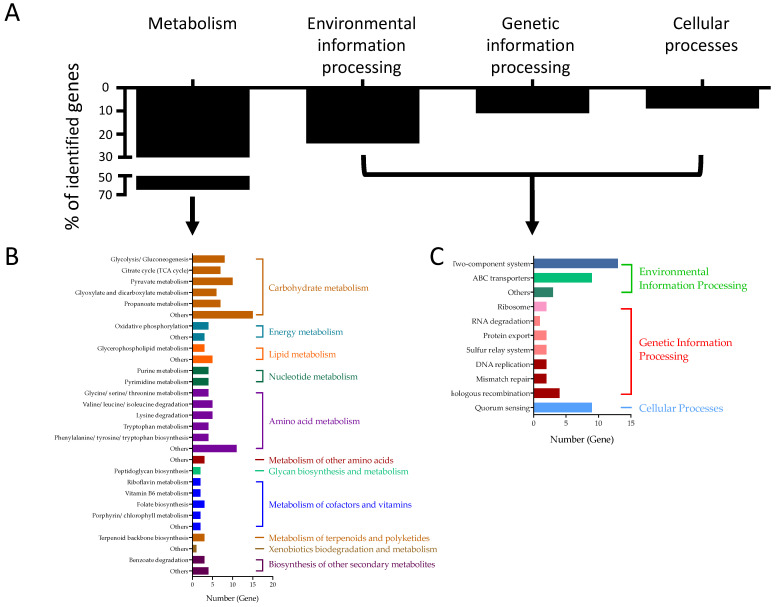
Kyoto Encyclopedia of Genes and Genomes (KEGG) enrichment analysis of the mutant strains significantly decreasing HMEC-1 damage rate: (**A**) genes were identified in the KEGG database and belonged to four major KEGG pathways; (**B**) the sub-pathway enrichment analysis of the genes in the metabolism pathway; and (**C**) the sub-pathway enrichment analysis of the genes in the other three pathways.

**Figure 3 antibiotics-11-00316-f003:**
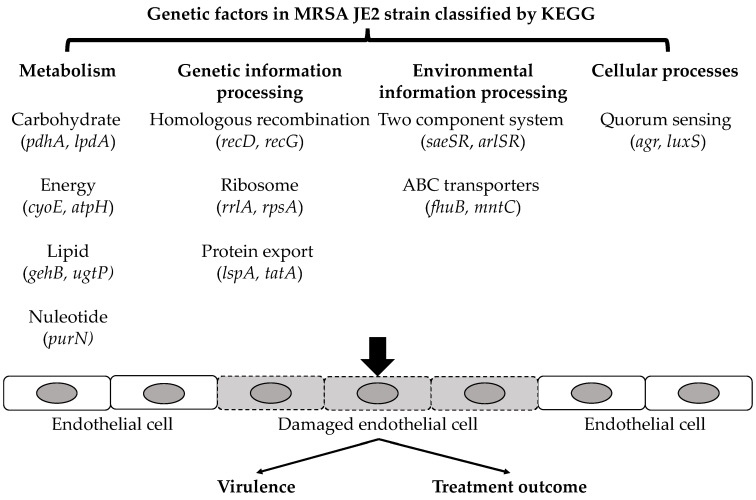
Genetic factors in MRSA JE2 strain contribute to the HMEC-1 damage by KEGG analysis. These factors may ultimately impact the pathogenesis and treatment outcome in MRSA endovascular infection.

**Table 1 antibiotics-11-00316-t001:** Mutants significantly decrease HMEC-1 damage vs. JE2 WT strain (EC damage rate ≤ 30%).

Locus	Gene Name	Description	% EC Damage (Mean ± SD)
SAUSA300_0261	hypothetical	conserved hypothetical protein	29.83 ± 8.34
SAUSA300_1172	hypothetical	M16 family peptidase	29.74 ± 4.80
SAUSA300_0083	hypothetical	hypothetical protein	29.70 ± 10.14
SAUSA300_1386	hypothetical	phiETA ORF59-like protein	29.57 ± 1.07
SAUSA300_0076	hypothetical	ABC transporter ATP-binding protein	29.57 ± 4.10
SAUSA300_1712	*ribH*	6,7-dimethyl-8-ribityllumazine synthase	29.49 ± 9.83
SAUSA300_1457	*malR*	maltose operon transcriptional repressor	29.46 ± 2.79
SAUSA300_1309	hypothetical	IS200 family transposase	29.41 ± 8.13
SAUSA300_1253	*glcT*	transcription antiterminator	29.37 ± 4.04
SAUSA300_1797	hypothetical	conserved hypothetical protein	29.37 ± 4.79
SAUSA300_1759	hypothetical	hypothetical protein	29.25 ± 2.85
SAUSA300_2386	hypothetical	beta-lactamase	29.13 ± 1.62
SAUSA300_2434	hypothetical	transporter protein	29.13 ± 5.28
SAUSA300_2037	hypothetical	ATP-dependent RNA helicase	28.67 ± 8.90
SAUSA300_1654	hypothetical	proline dipeptidase	28.46 ± 4.20
SAUSA300_0615	hypothetical	putative monovalent cation/H+ antiporter subunit F	28.45 ± 4.24
SAUSA300_1659	*tpx*	thiol peroxidase	28.41 ± 7.42
SAUSA300_1478	hypothetical	putative lipoprotein	28.28 ± 4.37
SAUSA300_2455	hypothetical	putative fructose-1,6-bisphosphatase	28.27 ± 5.83
SAUSA300_1297	*acyP*	acylphosphatase	28.23 ± 4.50
SAUSA300_2606	*hisF*	imidazole glycerol phosphate synthase subunit HisF	27.62 ± 4.01
SAUSA300_0795	hypothetical	hypothetical protein	27.38 ± 6.00
SAUSA300_1683	hypothetical	bifunctional 3-deoxy-7-phosphoheptulonate synthase/chorismate mutase	27.26 ± 6.86
SAUSA300_2618	hypothetical	hypothetical protein	27.23 ± 7.65
SAUSA300_1398	hypothetical	phiSLT ORF123-like protein	27.16 ± 11.43
SAUSA300_0059	hypothetical	conserved hypothetical protein	27.07 ± 7.67
SAUSA300_1764	*epiD*	lantibiotic epidermin biosynthesis protein EpiD	26.84 ± 3.46
SAUSA300_2332	hypothetical	heat shock protein	26.78 ± 8.46
SAUSA300_1040	hypothetical	hypothetical protein	26.74 ± 8.21
SAUSA300_2280	*fosB*	fosfomycin resistance protein FosB	26.67 ± 8.68
SAUSA300_1750	hypothetical	conserved hypothetical protein	26.62 ± 9.44
SAUSA300_0883	hypothetical	putative surface protein	26.40 ± 12.90
SAUSA300_1964	hypothetical	hypothetical protein	26.38 ± 7.19
SAUSA300_0290	hypothetical	putative lipoprotein	26.29 ± 8.56
SAUSA300_1672	*nagE*	phosphotransferase system, N-acetylglucosamine-specific IIBC component	26.21 ± 5.46
SAUSA300_2023	*rsbW*	anti-sigma-B factor, serine-protein kinase	26.01 ± 0.14
SAUSA300_0190	*ipdC*	indole-3-pyruvate decarboxylase	25.81 ± 7.93
SAUSA300_2413	hypothetical	hypothetical protein	25.79 ± 4.70
SAUSA300_0798	hypothetical	ABC transporter substrate-binding protein	25.59 ± 3.93
SAUSA300_0489	*ftsH*	putative cell division protein FtsH	25.55 ± 5.76
SAUSA300_1093	*pyrB*	aspartate carbamoyltransferase catalytic subunit	25.49 ± 1.23
SAUSA300_0517	hypothetical	RNA methyltransferase	25.39 ± 8.18
SAUSA300_1740	hypothetical	hypothetical protein	25.37 ± 9.05
SAUSA300_0540	hypothetical	HAD family hydrolase	25.26 ± 9.24
SAUSA300_2272	hypothetical	hypothetical protein	25.25 ± 4.80
SAUSA300_1968	hypothetical	putative phage transcriptional regulator	25.23 ± 9.97
SAUSA300_0642	hypothetical	hypothetical protein	25.21 ± 4.58
SAUSA300_2358	hypothetical	ABC transporter permease	25.11 ± 6.08
SAUSA300_1984	*mroQ*	hypothetical protein	25.07 ± 9.15
SAUSA300_1266	*trpF*	N-(5′-phosphoribosyl)anthranilate isomerase	25.05 ± 7.12
SAUSA300_2251	hypothetical	dehydrogenase family protein	25.00 ± 3.65
SAUSA300_0706	hypothetical	putative osmoprotectant ABC transporter ATP-binding protein	24.95 ± 11.00
SAUSA300_0941	hypothetical	putative ferrichrome ABC transporter	24.69 ± 6.43
SAUSA300_0951	*sspA*	V8 protease	24.55 ± 8.41
SAUSA300_1875	hypothetical	exonuclease	24.52 ± 10.68
SAUSA300_0566	hypothetical	amino acid permease	24.49 ± 5.06
SAUSA300_0871	hypothetical	hypothetical protein	24.49 ± 12.19
SAUSA300_0565	hypothetical	conserved hypothetical protein	24.43 ± 5.34
SAUSA300_0391	hypothetical	hypothetical protein	24.38 ± 0.45
SAUSA300_1328	hypothetical	putative drug transporter	24.10 ± 7.38
SAUSA300_2279	hypothetical	LysR family regulatory protein	23.92 ± 10.37
SAUSA300_0505	hypothetical	glutamine amidotransferase subunit PdxT	23.61 ± 3.46
SAUSA300_0470	*ksgA*	dimethyladenosine transferase	23.56 ± 7.13
SAUSA300_1106	hypothetical	putative lipoprotein	23.45 ± 8.92
SAUSA300_1991	*agrC*	accessory gene regulator protein C	23.44 ± 9.71
SAUSA300_0108	hypothetical	antigen, 67 kDa	23.33 ± 6.80
SAUSA300_2326	*araC*	transcription regulatory protein	23.30 ± 5.35
SAUSA300_1399	hypothetical	phiSLT ORF110-like protein	23.29 ± 0.65
SAUSA300_1942	hypothetical	hypothetical protein	23.29 ± 11.27
SAUSA300_0079	hypothetical	putative lipoprotein	23.27 ± 6.02
SAUSA300_1384	hypothetical	phiSLT ORF100b-like protein, holin	23.25 ± 6.98
SAUSA300_1950	hypothetical	hypothetical protein	23.24 ± 9.64
SAUSA300_0320	*gehB*	triacylglycerol lipase	23.13 ± 9.02
SAUSA300_0370	hypothetical	putative enterotoxin	23.06 ± 9.01
SAUSA300_1224	hypothetical	conserved hypothetical protein	22.85 ± 4.12
SAUSA300_1925	hypothetical	phiPVL ORF17-like protein	22.72 ± 9.85
SAUSA300_1271	hypothetical	hydrolase-like protein	22.57 ± 5.67
SAUSA300_0547	*sdrD*	sdrD protein	22.52 ± 1.23
SAUSA300_0561	hypothetical	hypothetical protein	22.37 ± 6.87
SAUSA300_2367	*hlgB*	gamma-hemolysin component B	22.27 ± 7.70
SAUSA300_1671	hypothetical	hypothetical protein	22.15 ± 10.08
SAUSA300_2341	*narJ*	respiratory nitrate reductase, subunit delta	22.11 ± 4.50
SAUSA300_0420	hypothetical	hypothetical protein	22.10 ± 8.19
SAUSA300_2281	*hutG*	formimidoylglutamase	22.05 ± 12.63
SAUSA300_1427	hypothetical	phiSLT ORF86-like protein	21.94 ± 2.49
SAUSA300_0691	*saeR*	DNA-binding response regulator SaeR	21.93 ± 10.56
SAUSA300_1519	hypothetical	hypothetical protein	21.86 ± 0.84
SAUSA300_0253	*scdA*	cell wall biosynthesis protein ScdA	21.83 ± 12.24
SAUSA300_2459	hypothetical	MarR family transcriptional regulator	21.58 ± 6.37
SAUSA300_2505	hypothetical	acetyltransferase	21.48 ± 5.28
SAUSA300_0652	hypothetical	hypothetical protein	21.46 ± 9.86
SAUSA300_1213	hypothetical	hypothetical protein	21.42 ± 8.18
SAUSA300_1216	hypothetical	cardiolipin synthetase	21.40 ± 13.46
SAUSA300_0395	hypothetical	superantigen-like protein	21.39 ± 9.28
SAUSA300_1016	*cyoE*	protoheme IX farnesyltransferase	21.38 ± 6.70
SAUSA300_1126	*rnc*	ribonuclease III	21.34 ± 5.04
SAUSA300_1437	hypothetical	phiSLT ORF204-like protein	21.26 ± 3.02
SAUSA300_2145	hypothetical	glycine betaine transporter	21.18 ± 9.85
SAUSA300_2288	hypothetical	ABC transporter ATP-binding protein	21.10 ± 15.49
SAUSA300_0698	*pabA*	para-aminobenzoate synthase, glutamine amidotransferase, component II	21.05 ± 4.75
SAUSA300_0519	hypothetical	hypothetical protein	20.86 ± 6.93
SAUSA300_2330	hypothetical	hypothetical protein	20.82 ± 4.02
SAUSA300_0141	*deoB*	phosphopentomutase	20.69 ± 9.71
SAUSA300_1684	hypothetical	hypothetical protein	20.53 ± 11.18
SAUSA300_1595	*tgt*	queuine tRNA-ribosyltransferase	20.53 ± 9.07
SAUSA300_0442	hypothetical	hypothetical protein	20.45 ± 3.70
SAUSA300_0744	*lgt*	prolipoprotein diacylglyceryl transferase	20.44 ± 5.61
SAUSA300_1576	*recD2*	helicase, RecD/TraA family	20.41 ± 6.63
SAUSA300_2088	*luxS*	S-ribosylhomocysteinase	20.40 ± 2.33
SAUSA300_0131	hypothetical	putative Bacterial sugar transferase	20.28 ± 13.49
SAUSA300_0649	hypothetical	hypothetical protein	20.23 ± 0.89
SAUSA300_2550	*nrdG*	anaerobic ribonucleotide reductase, small subunit	20.22 ± 10.12
SAUSA300_2168	hypothetical	hypothetical protein	20.16 ± 4.12
SAUSA300_2587	hypothetical	accessory secretory protein Asp1	20.06 ± 9.42
SAUSA300_2548	hypothetical	hypothetical protein	19.98 ± 7.37
SAUSA300_1021	hypothetical	hypothetical protein	19.92 ± 15.09
SAUSA300_0456	*rrlA*	23S ribosomal RNA	19.91 ± 0.15
SAUSA300_0431	hypothetical	hypothetical protein	19.86 ± 4.23
SAUSA300_1247	hypothetical	conserved hypothetical protein	19.79 ± 10.23
SAUSA300_2108	*mtlD*	mannitol-1-phosphate 5-dehydrogenase	19.74 ± 9.18
SAUSA300_2516	hypothetical	short chain dehydrogenase/reductase family oxidoreductase	19.65 ± 10.14
SAUSA300_0450	*treR*	trehalose operon repressor	19.59 ± 13.38
SAUSA300_0422	hypothetical	hypothetical protein	19.54 ± 2.66
SAUSA300_1739	hypothetical	hypothetical protein	19.47 ± 8.56
SAUSA300_0257	*lrgB*	antiholin-like protein LrgB	19.47 ± 17.61
SAUSA300_0056	hypothetical	hypothetical protein	19.05 ± 4.22
SAUSA300_2352	hypothetical	addiction module antitoxin	18.95 ± 11.82
SAUSA300_2236	hypothetical	hypothetical protein	18.82 ± 4.26
SAUSA300_1409	hypothetical	hypothetical protein	18.77 ± 11.78
SAUSA300_1304	hypothetical	hypothetical protein	18.73 ± 5.92
SAUSA300_1934	hypothetical	phi77 ORF020-like protein, phage major tail protein	18.68 ± 3.51
SAUSA300_1279	*phoU*	phosphate transport system regulatory protein PhoU	18.68 ± 7.74
SAUSA300_1217	hypothetical	ABC transporter ATP-binding protein	18.66 ± 8.42
SAUSA300_0468	hypothetical	TatD family hydrolase	18.62 ± 0.90
SAUSA300_2132	hypothetical	hypothetical protein	18.54 ± 17.28
SAUSA300_0288	*essD/esaD*	hypothetical protein	18.50 ± 12.03
SAUSA300_2461	hypothetical	glyoxalase family protein	18.38 ± 6.48
SAUSA300_1349	*bshA*	glycosyl transferase, group 1 family protein	18.26 ± 11.03
SAUSA300_1009	*typA*	GTP-binding protein	18.22 ± 6.42
SAUSA300_1755	*splD*	serine protease SplD	18.20 ± 6.01
SAUSA300_1966	hypothetical	phi77 ORF014-like protein, phage anti-repressor protein	18.04 ± 5.61
SAUSA300_1307	*arlS*	sensor histidine kinase protein	18.01 ± 7.14
SAUSA300_1918	*hlb*	truncated beta-hemolysin	17.91 ± 11.34
SAUSA300_1569	hypothetical	U32 family peptidase	17.90 ± 6.37
SAUSA300_1397	hypothetical	phiSLT ORF213-like protein, major tail protein	17.88 ± 16.40
SAUSA300_1032	hypothetical	putative iron compound ABC transporter iron compound-binding protein	17.87 ± 9.01
SAUSA300_0259	hypothetical	PTS system, IIA component	17.72 ± 4.08
SAUSA300_1070	hypothetical	hypothetical protein	17.66 ± 6.61
SAUSA300_1474	hypothetical	hypothetical protein	17.57 ± 3.84
SAUSA300_1451	hypothetical	short chain dehydrogenase/reductase family oxidoreductase	17.47 ± 4.46
SAUSA300_0769	hypothetical	hypothetical protein	17.42 ± 7.43
SAUSA300_2098	*arsR*	ArsR family transcriptional regulator	17.36 ± 8.42
SAUSA300_0094	hypothetical	hypothetical protein	17.32 ± 9.77
SAUSA300_1470	*ispA*	geranyltranstransferase	17.29 ± 13.19
SAUSA300_1403	hypothetical	phiSLT ORF412-like protein, portal protein	17.28 ± 10.80
SAUSA300_2432	hypothetical	MutT/NUDIX family hydrolase	17.26 ± 15.82
SAUSA300_0631	hypothetical	putative nucleoside transporter	17.25 ± 11.20
SAUSA300_1000	*potB*	spermidine/putrescine ABC transporter permease	17.14 ± 5.86
SAUSA300_2559	hypothetical	DNA-binding response regulator	17.10 ± 8.85
SAUSA300_2467	*srtA*	sortase	17.01 ± 6.72
SAUSA300_2300	hypothetical	transcriptional regulator, TetR family	16.92 ± 5.04
SAUSA300_0916	hypothetical	hypothetical protein	16.89 ± 2.85
SAUSA300_1444	*scpB*	segregation and condensation protein B	16.85 ± 6.40
SAUSA300_0995	hypothetical	branched-chain alpha-keto acid dehydrogenase subunit E2	16.83 ± 18.68
SAUSA300_0419	hypothetical	tandem lipoprotein	16.78 ± 3.58
SAUSA300_1563	*accC*	acetyl-CoA carboxylase, biotin carboxylase	16.73 ± 11.04
SAUSA300_2027	*alr*	alanine racemase	16.70 ± 16.05
SAUSA300_2607	*hisA*	phoribosyl)-5-((5-phosphoribosylamino)methylideneamino) imidazole-4-carboxamide	16.70 ± 11.46
SAUSA300_0023	hypothetical	hypothetical protein	16.69 ± 16.09
SAUSA300_1622	*tig*	trigger factor	16.44 ± 5.67
SAUSA300_0011	hypothetical	hypothetical protein	16.37 ± 4.02
SAUSA300_1097	*pyrF*	orotidine 5′-phosphate decarboxylase	16.34 ± 8.94
SAUSA300_1339	hypothetical	hypothetical protein	16.25 ± 5.49
SAUSA300_0585	hypothetical	hypothetical protein	16.24 ± 13.38
SAUSA300_0839	*nfu*	hypothetical protein	16.23 ± 12.30
SAUSA300_0071	hypothetical	ISSep1-like transposase	16.19 ± 3.17
SAUSA300_0651	hypothetical	CHAP domain-contain protein	16.09 ± 6.91
SAUSA300_1599	hypothetical	hypothetical protein	16.02 ± 7.75
SAUSA300_1607	hypothetical	hypothetical protein	16.02 ± 8.76
SAUSA300_0588	hypothetical	hypothetical protein	15.86 ± 15.72
SAUSA300_2276	hypothetical	peptidase, M20/M25/M40 family	15.84 ± 1.33
SAUSA300_2055	*murA*	UDP-N-acetylglucosamine 1-carboxyvinyltransferase	15.79 ± 10.49
SAUSA300_0808	hypothetical	hypothetical protein	15.69 ± 12.88
SAUSA300_0759	*gpmI*	phosphoglyceromutase	15.68 ± 9.84
SAUSA300_0857	*ppiB*	hypothetical protein	15.66 ± 4.76
SAUSA300_1051	hypothetical	hypothetical protein	15.51 ± 14.05
SAUSA300_1383	hypothetical	phiSLT ORF484-like protein, lysin	15.46 ± 15.13
SAUSA300_1566	hypothetical	hypothetical protein	15.42 ± 14.25
SAUSA300_2040	hypothetical	hypothetical protein	15.42 ± 12.63
SAUSA300_1145	*xerC*	tyrosine recombinase xerC	15.33 ± 4.57
SAUSA300_0687	hypothetical	putative hemolysin	15.14 ± 12.23
SAUSA300_0630	hypothetical	ABC transporter ATP-binding protein	15.07 ± 10.45
SAUSA300_1577	hypothetical	TPR domain-containing protein	14.93 ± 1.75
SAUSA300_1288	*dapA*	dihydrodipicolinate synthase	14.75 ± 7.53
SAUSA300_1937	hypothetical	phi77 ORF045-like protein	14.69 ± 8.83
SAUSA300_1419	hypothetical	phiSLT ORF80-like protein	14.65 ± 9.06
SAUSA300_2345	*nirD*	nitrite reductase (NAD(P)H), small subunit	14.54 ± 4.64
SAUSA300_1365	*rpsA*	30S ribosomal protein S1	14.53 ± 3.46
SAUSA300_0029	hypothetical	hypothetical protein	14.39 ± 3.30
SAUSA300_2575	hypothetical	BglG family transcriptional antiterminator	14.12 ± 4.67
SAUSA300_1497	hypothetical	glycine dehydrogenase subunit 1	14.08 ± 4.09
SAUSA300_1682	*ccpA*	catabolite control protein A	14.04 ± 8.43
SAUSA300_0657	hypothetical	hypothetical protein	14.02 ± 7.45
SAUSA300_1955	hypothetical	putative endodeoxyribonuclease RusA	13.92 ± 10.12
SAUSA300_0924	*ktrD*	sodium transport family protein	13.85 ± 14.78
SAUSA300_0077	hypothetical	ABC transporter ATP-binding protein	13.80 ± 6.67
SAUSA300_0504	*pdxS*	pyridoxal biosynthesis lyase PdxS	13.58 ± 7.70
SAUSA300_0195	hypothetical	transcriptional regulator	13.06 ± 13.37
SAUSA300_1308	*arlR*	DNA-binding response regulator	13.05 ± 5.02
SAUSA300_0859	hypothetical	NADH-dependent flavin oxidoreductase	12.99 ± 7.37
SAUSA300_1721	hypothetical	hypothetical protein	12.97 ± 3.93
SAUSA300_0186	*argC*	N-acetyl-gamma-glutamyl-phosphate reductase	12.92 ± 16.00
SAUSA300_2641	hypothetical	hypothetical protein	12.90 ± 8.36
SAUSA300_0987	hypothetical	cytochrome D ubiquinol oxidase, subunit II	12.85 ± 10.22
SAUSA300_1696	*dat*	D-alanine aminotransferase	12.74 ± 5.48
SAUSA300_1283	hypothetical	phosphate ABC transporter, phosphate-binding protein PstS	12.73 ± 9.23
SAUSA300_1185	*miaB*	(dimethylallyl)adenosine tRNA methylthiotransferase	12.62 ± 10.40
SAUSA300_2365	*hlgA*	gamma-hemolysin component A	12.56 ± 10.54
SAUSA300_1394	hypothetical	hypothetical protein	12.34 ± 12.26
SAUSA300_0115	*sirC*	iron compound ABC transporter permease SirC	12.30 ± 6.17
SAUSA300_2284	hypothetical	hypothetical protein	12.20 ± 10.36
SAUSA300_2225	*moaC*	molybdenum cofactor biosynthesis protein MoaC	12.08 ± 9.05
SAUSA300_0244	hypothetical	zinc-binding dehydrogenase family oxidoreductase	12.05 ± 9.79
SAUSA300_2022	*rpoF*	RNA polymerase sigma factor SigB	12.05 ± 6.83
SAUSA300_1089	*lspA*	lipoprotein signal peptidase	11.97 ± 6.81
SAUSA300_1618	*hemX*	hemA concentration negative effector hemX	11.88 ± 1.05
SAUSA300_0117	*sirA*	iron compound ABC transporter iron compound-binding protein SirA	11.83 ± 7.84
SAUSA300_0899	*mecA*	adaptor protein	11.58 ± 10.37
SAUSA300_2492	hypothetical	acetyltransferase family protein	11.55 ± 7.80
SAUSA300_1433	hypothetical	putative phage regulatory protein	11.41 ± 8.17
SAUSA300_1244	*mscL*	large conductance mechanosensitive channel protein	11.32 ± 7.21
SAUSA300_0049	hypothetical	hypothetical protein	11.30 ± 0.62
SAUSA300_1667	hypothetical	putative glycerophosphoryl diester phosphodiesterase	11.30 ± 7.51
SAUSA300_0994	*pdhB*	pyruvate dehydrogenase E1 component, beta subunit	11.20 ± 8.12
SAUSA300_0974	*purN*	phosphoribosylglycinamide formyltransferase	11.07 ± 8.08
SAUSA300_0067	hypothetical	universal stress protein	11.02 ± 9.02
SAUSA300_1590	*rsh (relA)*	GTP pyrophosphokinase	10.95 ± 7.18
SAUSA300_0526	hypothetical	methyltransferase small subunit	10.80 ± 10.78
SAUSA300_0952	hypothetical	aminotransferase, class I	10.57 ± 6.79
SAUSA300_1694	*trmB*	tRNA (guanine-N(7)-)-methyltransferase	10.55 ± 16.08
SAUSA300_0041	hypothetical	hypothetical protein	10.41 ± 2.09
SAUSA300_1449	hypothetical	MutT/nudix family protein	10.11 ± 13.24
SAUSA300_0724	hypothetical	hypothetical protein	10.06 ± 2.60
SAUSA300_1757	*splB*	serine protease SplB	9.41 ± 4.17
SAUSA300_0476	hypothetical	hypothetical protein	9.18 ± 8.05
SAUSA300_2052	hypothetical	single-stranded DNA- binding protein family	9.11 ± 18.19
SAUSA300_2176	*cbiO*	cobalt transporter ATP-binding subunit	9.03 ± 9.11
SAUSA300_1112	*stp1*	protein phosphatase 2C domain-containing protein	8.98 ± 14.19
SAUSA300_0789	hypothetical	putative thioredoxin	8.89 ± 18.33
SAUSA300_0379	*ahpF*	alkyl hydroperoxide reductase subunit F	8.46 ± 4.49
SAUSA300_0348	*tatA*	twin arginine-targeting protein translocase	8.36 ± 5.53
SAUSA300_0469	*rnmV*	hypothetical protein	8.35 ± 0.35
SAUSA300_1792	hypothetical	hypothetical protein	8.20 ± 4.58
SAUSA300_2061	*atpH*	F0F1 ATP synthase subunit delta	7.98 ± 1.29
SAUSA300_1092	*pyrP*	uracil permease	7.85 ± 2.60
SAUSA300_0905	hypothetical	hypothetical protein	7.61 ± 3.76
SAUSA300_0444	*gltC*	LysR family regulatory protein	7.59 ± 2.70
SAUSA300_2646	*trmE*	tRNA modification GTPase TrmE	7.41 ± 8.81
SAUSA300_2105	*mtlF*	PTS system, mannitol specific IIBC component	6.95 ± 0.84
SAUSA300_2486	*clpL*	putative ATP-dependent Clp proteinase	6.73 ± 0.02
SAUSA300_1887	*pcrB*	geranylgeranylglyceryl phosphate synthase-like protein	6.58 ± 3.46
SAUSA300_1653	hypothetical	metal-dependent hydrolase	6.25 ± 8.63
SAUSA300_2393	*opuCa*	glycine betaine/carnitine/choline ABC transporter ATP-binding protein	6.25 ± 7.87
SAUSA300_1183	hypothetical	2-oxoglutarate ferredoxin oxidoreductase subunit beta	6.19 ± 1.88
SAUSA300_0393	hypothetical	hypothetical protein	6.18 ± 2.30
SAUSA300_0174	hypothetical	hypothetical protein	6.15 ± 1.39
SAUSA300_0841	hypothetical	hypothetical protein	5.97 ± 2.99
SAUSA300_1096	*carB*	carbamoyl phosphate synthase large subunit	5.89 ± 2.89
SAUSA300_2593	hypothetical	hypothetical protein	5.84 ± 3.04
SAUSA300_0221	*pflA*	pyruvate formate-lyase activating enzyme	5.68 ± 18.96
SAUSA300_0996	*lpdA*	dihydrolipoamide dehydrogenase	5.49 ± 2.87
SAUSA300_1992	*agrA*	accessory gene regulator protein A	5.34 ± 14.81
SAUSA300_1147	*hslU*	ATP-dependent protease ATP-binding subunit HslU	4.99 ± 6.72
SAUSA300_1120	*recG*	ATP-dependent DNA helicase RecG	4.60 ± 0.15
SAUSA300_2078	*murA*	UDP-N-acetylglucosamine 1-carboxyvinyltransferase	3.18 ± 3.15
SAUSA300_1583	*cymR*	hypothetical protein	2.48 ± 0.46
SAUSA300_0992	hypothetical	hypothetical protein	2.24 ± 20.30
SAUSA300_0634	*fhuB*	ferrichrome transport permease fhuB	2.22 ± 4.57
SAUSA300_0750	*whiA*	hypothetical protein	1.88 ± 4.32
SAUSA300_2485	hypothetical	methylated DNA-protein cysteine methyltransferase	1.78 ± 9.18
SAUSA300_0426	hypothetical	hypothetical protein	0.95 ± 5.09
SAUSA300_2598	*capA*	capsular polysaccharide biosynthesis protein Cap1A	0.85 ± 1.11
SAUSA300_2246	hypothetical	hypothetical protein	0.51 ± 16.59
SAUSA300_2518	hypothetical	hydrolase family protein	0.42 ± 7.59
SAUSA300_0355	hypothetical	acetyl-CoA acetyltransferase	−0.68 ^a^ ± 4.44
SAUSA300_0398	hypothetical	superantigen-like protein	−0.83 ^a^ ± 2.42
SAUSA300_2226	*moaB*	molybdenum cofactor biosynthesis protein B	−1.15 ^a^ ± 5.09
SAUSA300_0945	hypothetical	isochorismate synthase family protein	−1.17 ^a^ ± 14.05
SAUSA300_0904	*yjbI*	hypothetical protein	−1.32 ^a^ ± 9.61
SAUSA300_0423	hypothetical	hypothetical protein	−2.20 ^a^ ± 9.08
SAUSA300_1422	hypothetical	phiSLT ORF65-like protein	−2.77 ^a^ ± 6.35
SAUSA300_0068	hypothetical	cadmium-exporting ATPase, truncation	−2.79 ^a^ ± 8.85
SAUSA300_1870	hypothetical	hypothetical protein	−2.92 ^a^ ± 15.58
SAUSA300_1139	*sucD*	succinyl-CoA synthetase subunit alpha	−2.94 ^a^ ± 8.32
SAUSA300_0918	*ugtP*	diacylglycerol glucosyltransferase	−3.09 ^a^ ± 8.63
SAUSA300_0597	hypothetical	putative endonuclease III	−3.15 ^a^ ± 14.78
SAUSA300_0326	hypothetical	hypothetical protein	−3.64 ^a^ ± 2.40
SAUSA300_0690	*saeS*	sensor histidine kinase SaeS	−4.88 ^a^ ± 14.01
SAUSA300_0560	*vraB*	acetyl-CoA c-acetyltransferase	−5.06 ^a^ ± 6.53
SAUSA300_2334	hypothetical	hypothetical protein	-5.12 ^a^ ± 7.55
SAUSA300_2025	*rsbU*	sigma-B regulation protein	−5.19 ^a^ ± 6.08
SAUSA300_2152	*lacD*	tagatose 1,6-diphosphate aldolase	−5.59 ^a^ ± 11.59
SAUSA300_1680	*acuA*	acetoin utilization protein AcuA	−5.94 ^a^ ± 10.87
SAUSA300_2024	*rsbV*	anti-sigma-B factor, antagonist	−6.77 ^a^ ± 14.71
SAUSA300_0618	*mntC*	ABC transporter substrate-binding protein	−6.85 ^a^ ± 4.61
SAUSA300_1876	hypothetical	DNA polymerase IV	−6.91 ^a^ ± 9.59
SAUSA300_1465	hypothetical	2-oxoisovalerate dehydrogenase, E1 component, beta subunit	−7.15 ^a^ ± 6.73
SAUSA300_1573	hypothetical	Holliday junction resolvase-like protein	−10.10 ^a^ ± 6.88
SAUSA300_1473	*nusB*	transcription antitermination protein NusB	−10.84 ^a^ ± 10.00
SAUSA300_1357	*aroC*	chorismate synthase	−11.88 ^a^ ± 0.89
SAUSA300_1095	*carA*	carbamoyl phosphate synthase small subunit	−14.12 ^a^ ± 10.52
SAUSA300_1469	*argR*	arginine repressor	−14.16 ^a^ ± 8.61
SAUSA300_1615	*hemB*	delta-aminolevulinic acid dehydratase	−14.95 ^a^ ± 14.12
SAUSA300_1467	*lpdA*	dihydrolipoamide dehydrogenase	−15.68 ^a^ ± 14.07
SAUSA300_0993	*pdhA*	pyruvate dehydrogenase E1 component, alpha subunit	−17.05 ^a^ ± 10.66
SAUSA300_0752	*clpP*	ATP-dependent Clp protease proteolytic subunit	−17.66 ^a^ ± 11.34
SAUSA300_1715	*ribD*	riboflavin biosynthesis protein	−23.78 ^a^ ± 4.28

^a^ EC damage below zero is due to the A_560nm_ value of the mutant was higher than the A_560nm_ of the negative control.

**Table 2 antibiotics-11-00316-t002:** Mutants significantly increase HMEC-1 damage vs. JE2 WT strain (EC damage rate ≥ 60%).

Locus	Gene Name	Description	% EC Damage (Mean ± SD)
SAUSA300_1197	ND ^a^	glutathione peroxidase	62.86 ± 5.67
SAUSA300_1333	hypothetical	conserved hypothetical protein	62.17 ± 3.05
SAUSA300_1485	hypothetical	conserved hypothetical protein	61.86 ± 6.12
SAUSA300_2221	*moaD*	molybdopterin converting factor, subunit 1	61.64 ± 3.61
SAUSA300_0206	*azoR*	flavodoxin family protein	60.82 ± 6.24
SAUSA300_0335	*mepA*	MATE efflux family protein	60.15 ± 8.13

^a^ ND: not determined.

**Table 3 antibiotics-11-00316-t003:** Verification of EC damage of JE WT strain and selected mutants using 24-well plates assay.

Locus	Group	Gene Name	% EC Damage (Mean ± SD)	
			384-Well Plates	24-Well Plates
JE2	Wildtype		46.19 ± 2.97	42.43 ± 6.44
SAUSA300_1197	EC damage ≥ 60% in 384-well plates	hypothetical	62.86 ± 5.67	59.40 ± 1.50
SAUSA300_1333		hypothetical	62.17 ± 3.05	66.92 ± 0.84
SAUSA300_1485		hypothetical	61.86 ± 6.12	61.75 ^a^
SAUSA300_2221		*moaD*	61.64 ± 3.61	59.90 ± 1.08
SAUSA300_0206		hypothetical	60.82 ± 6.24	69.33 ± 0.48
SAUSA300_0335		hypothetical	60.15 ± 8.31	63.35 ± 2.06
SAUSA300_1040	EC damage ≤ 30% in 384-well plates	hypothetical	26.74 ± 8.21	30.92 ^a^
SAUSA300_1875		hypothetical	24.52 ± 10.68	30.51 ^a^
SAUSA300_0871		hypothetical	24.49 ± 12.19	28.60 ^a^
SAUSA300_1950		hypothetical	23.24 ± 9.64	25.87 ^a^
SAUSA300_0253		*scdA*	21.83 ± 12.24	22.52 ^a^
SAUSA300_0649		hypothetical	20.24 ± 0.89	22.65 ^a^
SAUSA300_2587		hypothetical	20.06 ± 9.42	26.45 ^a^
SAUSA300_0631		hypothetical	17.25 ± 11.20	23.00 ^a^
SAUSA300_2027		*alr*	16.70 ± 16.05	3.28 ± 1.38
SAUSA300_2055		*murA*	15.79 ± 10.49	7.62 ± 0.59
SAUSA300_1682		*ccpA*	14.04 ± 8.43	13.43 ^a^
SAUSA300_1696		*dat*	12.74 ± 5.48	14.99 ± 1.34
SAUSA300_0974		*purN*	11.07 ± 8.08	20.58 ^a^
SAUSA300_1563		*accC*	16.73 ± 11.04	11.82 ± 0.72
SAUSA300_0041		hypothetical	10.41 ± 2.09	3.30 ^a^
SAUSA300_0994		*pdhB*	11.20 ± 8.12	19.36 ^a^
SAUSA300_0186		*argC*	12.92 ± 16.00	15.20 ± 2.13
SAUSA300_1992		*agrA*	5.34 ± 14.81	−3.82 ± 1.77
SAUSA300_0355		hypothetical	−0.68 ± 4.44	−1.20 ^a^
SAUSA300_0690		*saeS*	−4.89 ± 14.01	−12.80 ± 1.77

^a^ Verification of these mutants was performed once using the 24-well plates assay.

**Table 4 antibiotics-11-00316-t004:** Numbers of genes from different KEGG pathway categories.

Categories	Sub-Groups	No. of Mutants with Decreased HMEC-1 Damage	No. of Mutants with Increased HMEC-1 Damage
Metabolism	Carbohydrate metabolism	53	
	Amino acid metabolism	33	
	Metabolism of cofactors and vitamins	11	
	Lipid metabolism	8	1
	Nucleotide metabolism	8	
	Biosynthesis of other secondary metabolites	7	
	Energy metabolism	7	
	Metabolism of other amino acids	3	1
	Metabolism of terpenoids and polyketides	3	
	Glycan biosynthesis and metabolism	2	
	Xenobiotics biodegradation and metabolism	1	
Genetic information processing	Homologous recombination	4	
	DNA replication	2	
	Mismatch repair	2	
	Protein export	2	
	Ribosome	2	
	Sulfur relay system	2	1
	RNA degradation	1	
Environmental information processing	Two-component system	13	
	ABC transporters	9	
	Other	3	
Cellular processes	Quorum sensing	9	
Total		185	3

## Data Availability

Not applicable.

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
