# Peer review of "Identification of Methicillin-Resistant Staphylococcus aureus (MRSA) Genetic Factors Involved in Human Endothelial Cells Damage, an Important Phenotype Correlated with Persistent Endovascular Infection"

_antibiotics, 2022, doi:10.3390/antibiotics11030316_

Round 1

Reviewer 1 Report

Xiao et al report in this manuscript the identification of genetic factors affecting human endothelial cell damage by screening the Nebraska transposon mutant library. They showed mutations in more than 300 genes had significant impact on EC damage. This is a straightforward study. The results will provide useful information for future studies on the pathogenesis of S. aureus endovascular infections. I have only some minor suggestions. 

1. Line 77-81: It is unclear whether all these genes or just the arlR, was tested. Also, in line 252, arlR mutation was stated to have >70% effect (ref 14) but here it was reported to have only >30% effect. Please discuss the discrepancy. 

2. Table 1 is the heart of this report. It would be much more informative if the % damage of each mutation is included in the Table. 

3. Fig. 3 could be incorporated into Fig. 2A as these figures are essentially the same. 

4. Line 141: determiners

5. Method 4.3: it is unclear whether the method as described is for 384-well or 96-well format. 

6. Verification of the screening was described in Method 4.4 but the results were not shown. The experiments are important for verifying the high-throughput screening and the results should be presented. 

Author Response

Response to Reviewer 1

  1. Line 77-81: It is unclear whether all these genes or just the arlR, was tested. Also, in line 252, arlR mutation was stated to have >70% effect (ref 14) but here it was reported to have only >30%effect. Please discuss the discrepancy.

In the current study, all the mutant strains in the Nebraska Transposon Mutant Library, including the genes described in lines 79-81, were tested. In addition, we demonstrated that the arlR mutant and JE2 WT strains caused 13.05 ± 5.02% and 46.2 ± 3.0% EC damage, respectively. These results indicate that the arlR mutant had ~70% EC damage reduction as compared to the JE2 WT strain, which is consistent with previously reported data (Ref#13).

  1. Table 1 is the heart of this report. It would be much more informative if the % damage of each mutation is included in the Table.

This suggestion is well taken.

  1. Fig. 3 could be incorporated into Fig. 2A as these figures are essentially the same.

Thanks for your suggestion. Figure 2 presents a general classification of genetic factors impacting the EC damage by KEGG enrichment analysis, while Figure 3 shows some specific virulence factors in each class that are known to play roles in MRSA virulence and treatment outcome. Therefore, we prefer to keep these two figures as the original submission.

  1. Line 141: determiners

Thank you for your careful review. We have corrected this typo (see line 139).

  1. Method 4.3: it is unclear whether the method as described is for 384-well or 96-well format.

Thanks for raising this point. The method as described in Method 4.3 is for 384-well and 24-well format. We have specified this fact in line 240.

  1. Verification of the screening was described in Method 4.4 but the results were not shown. The experiments are important for verifying the high-throughput screening and the results should be presented.

This suggestion is well taken. We have presented the verification results in Table 3 and described them in lines 95-96.

Reviewer 2 Report

This manuscript is a study of a comprehensive screening of genetic factors affecting human EC damage in Staphylococcus aureus using a mutants library.
The paper is well-written, and the authors have clearly worked hard to produce a comprehensive dataset and detailed description of their methods.
There are no particular methodological problems with this paper, and I have no hesitation in recommending it for publication with some minor modifications, but Figure 3 is a bit difficult to understand and may need some revisions.

Author Response

Response to Reviewer 2

This manuscript is a study of a comprehensive screening of genetic factors affecting human EC damage in Staphylococcus aureus using a mutants library. The paper is well-written, and the authors have clearly worked hard to produce a comprehensive dataset and detailed description of their methods. There are no particular methodological problems with this paper, and I have no hesitation in recommending it for publication with some minor modifications, but Figure 3 is a bit difficult to understand and may need some revisions.

We greatly appreciate the reviewer’s positive review.

Reviewer 3 Report

In the current manuscript Authors presented a continuation of research on the characteristics and genetic determinants of the MRSA S. aureus ability to destruct endothelial cells. The new data is presented in detail, but I have some comments on the manuscript.

In the discussion I do not find significant and convincing reference to the second part of the title, the conclusions to this statement should be more emphasized or the title should be revised.

The results described in point 4.4. "Verification of the HMEC-1 Damage Screening Results" should be placed in the Results section in the manuscript and other results from mutant screening and presented in Figure 1 should be attached to the manuscript in the form of a supplementary data.

Minor remarks:

double numbering in the reference list

no italics in the name of the species in the list of literature 

Author Response

Response to Reviewer 3

In the current manuscript Authors presented a continuation of research on the characteristics and genetic determinants of the MRSA S. aureus ability to destruct endothelial cells. The new data is presented in detail, but I have some comments on the manuscript.

1. In the discussion I do not find significant and convincing reference to the second part of the title, the conclusions to this statement should be more emphasized or the title should be revised. The results described in point 4.4. "Verification of the HMEC-1 Damage Screening Results" should be placed in the Results section in the manuscript and other results from mutant screening and presented in Figure 1 should be attached to the manuscript in the form of a supplementary data.

Thanks for the great comments. We addressed the second part of the title related to the relationship between in vitro EC damage and persistent endovascular infection due to MRSA in the revised Discussion section (lines 131-136). In addition, more references have been added to support this connection.

As suggested, we present the verification data of the HMEC-1 damage screening (384-wall vs 24-wall) on selected strains in a new table (Table 3) and described in the revised Results 2.1 section (lines 95-96).

The HMEC-1 damage data of the rest mutant strain in the Nebraska Transposon Mutant Library is presented in Table S1 in the Supplementary Materials and reported in lines 94-95.

2. Minor remarks:

double numbering in the reference list no italics in the name of the species in the list of literature

Thank you for your careful review. We have corrected the errors in the revised References section.

Round 2

Reviewer 3 Report

Authors made corrections as per comments.